# Acyl carrier protein promotes MukBEF action in *Escherichia coli* chromosome organization-segregation

Josh P. Prince [1,4], Jani R. Bolla [2,3,5], Gemma L. M. Fisher [1,6], Jarno Mäkelä [1,7], Marjorie Fournier[1], Carol V. Robinson [2,3], Lidia K. Arciszewska[1] & David J. Sherratt [1✉]

Structural Maintenance of Chromosomes (SMC) complexes act ubiquitously to compact DNA linearly, thereby facilitating chromosome organization-segregation. SMC proteins have a conserved architecture, with a dimerization hinge and an ATPase head domain separated by a long antiparallel intramolecular coiled-coil. Dimeric SMC proteins interact with essential accessory proteins, kleisins that bridge the two subunits of an SMC dimer, and HAWK/KITE proteins that interact with kleisins. The ATPase activity of the *Escherichia coli* SMC protein, MukB, which is essential for its in vivo function, requires its interaction with the dimeric kleisin, MukF that in turn interacts with the KITE protein, MukE. Here we demonstrate that, in addition, MukB interacts specifically with Acyl Carrier Protein (AcpP) that has essential functions in fatty acid synthesis. We characterize the AcpP interaction at the joint of the MukB coiled-coil and show that the interaction is necessary for MukB ATPase and for MukBEF function in vivo.

[1] Department of Biochemistry, University of Oxford, South Parks Road, Oxford OX1 3QU, UK. [2] Physical and Theoretical Chemistry Laboratory, University of Oxford, South Parks Road, Oxford OX1 3QZ, UK. [3] The Kavli Institute for Nanoscience Discovery, South Parks Road, Oxford OX1 3QU, UK. [4] Present address: Meiosis Group, Medical Research Council London Institute of Medical Sciences, Du Cane Road, London W12 0NN, UK. [5] Present address: Department of Plant Sciences, University of Oxford, South Parks Road, Oxford OX1 3QU, UK. [6] Present address: DNA Motors Group, Medical Research Council London Institute of Medical Sciences, Du Cane Road, London W12 0NN, UK. [7] Present address: ChEM-H Institute, Stanford University, 290 Jane Stanford Way, Stanford, CA 94305, USA. ✉email: david.sherratt@bioch.ox.ac.uk

SMC complexes act by linearly compacting chromosomal DNA, to organize chromosomes and facilitate their segregation prior to cell division, through individualizing newly replicated sister chromosomes[1–3]. In *Escherichia coli*, the SMC complex, MukBEF, is composed of three proteins, the SMC protein MukB, the kleisin, MukF and the KITE protein, MukE[4–6]. Although divergent in primary sequence from other SMC proteins, MukB shares common ancestral and architectural features including an ABC-like ATPase head domain, a ~50 nm long antiparallel coiled-coil and a dimerization hinge domain (Fig. 1a). In addition, MukB retains two highly conserved discontinuities within the coiled-coils. The first, the "joint", located ~100 amino acids from the head domain, is highly conserved between SMC complexes, and has been suggested to aid flexibility for head engagement during ATP hydrolysis cycles[2,7–10]. The other, roughly halfway along the coiled-coils, the "elbow", enables the protein to fold upon itself bringing the hinge domain in close proximity to one of the two ATPase heads, though the functional implications of this are unclear[8,11–13]. As with other SMC proteins, MukB dimers interact with their kleisin, MukF, through two distinct interaction sites; one in the "neck" region of the coiled-coils, located between the head and the joint of one monomer, and the "cap" region of the partner ATPase head (Fig. 1a)[5,14,15]. Unusually among kleisins, MukF dimerizes through an additional N-terminal dimeric winged-helix domain (WHD). This enables the joining of two dimeric MukBEF complexes into dimer of dimer (DoD) complexes that are essential for in vivo MukBEF function[6,15,16]. MukE dimers interact with MukF; thus the complete MukBEF complex has a 4:4:2 B:E:F stoichiometry[6,16]. MukB ATP hydrolysis results from the engagement of two head domains that create two shared ATP binding sites. MukB alone has minimal ATPase activity, but is activated in the presence of MukF and further modulated by the interactions with MukE and DNA[14]. ATPase activity is essential for in vivo function, as mutant MukB proteins that are deficient in ATP hydrolysis (MukB[E1407Q], hereafter referred to as MukB[EQ]) or ATP binding (MukB[D1406A], hereafter referred to as MukB[DA]) display $\Delta mukB$ phenotypes[6,17,18]. Cells lacking functional MukBEF, grown in a minimal medium under permissive conditions, have disorganized chromosomes with misplaced genetic loci, and fail to properly segregate sister chromosomes, leading to anucleate cell production[4,6,18–20]. Furthermore, $\Delta mukB$ cells are inviable during rapid growth that supports overlapping replication cycles, apparently because of failures in chromosome segregation[4,6,18,20]. Because the lack of functional MukB can result in reduced proliferation in organisms that cause disease phenotypes, such as *Vibrio vulnificus*[21], MukB is a potential target for new antimicrobial therapeutics[22].

Acyl Carrier Protein (AcpP) has been repeatedly reported to co-purify with MukB[16,23–25]. Since AcpP is a highly abundant *E. coli* protein (the most recent estimates give a range of $86–358 \times 10^3$ molecules/cell, dependent on growth medium-dependent cell size [~100 μM]; >100 times molar excess over endogenous MukB)[6,26], it was not clear from early reports whether this reflected a specific interaction or a fortuitous association. AcpP is an essential hub protein that through a covalent interaction with its phosphopantetheine (PPant) arm, shuttles intermediates along the fatty acid biosynthesis pathway by a series of acyl transfer reactions (Fig. 1a) (reviewed in ref. [27]). In addition, AcpP has been shown to interact with other unrelated protein partners including SpoT, IscS, and SecA[28–30]. Searches for binding partners of AcpP have also indicated an interaction with MukB, although any functional significance to this interaction was not explored[28,30,31].

Here, we identify the AcpP binding site on MukB and analyze the functional consequences of this interaction in vitro and in vivo. We show that the interaction of AcpP with a conserved region of the coiled-coils, in the MukB joint, is essential for MukB ATPase activity. MukBEF complexes of wild-type stoichiometry assemble in the absence of AcpP. The binding of AcpP to MukB inhibits higher-order intermolecular coiled-coil interactions between MukB molecules in vitro, consistent with a hypothesis in which AcpP binding to the MukBEF joint facilitates essential conformational changes in the complexes during cycles of ATP binding and hydrolysis. Mutations within the MukB–AcpP binding site reduce AcpP association and thus impair MukB ATPase activity. Importantly, these mutations result in an altered pattern of MukBEF complex localization within cells, including an increased association with the replication termination region (*ter*), consistent with the impaired ATPase function. We propose that AcpP is an essential partner for MukBEF action in chromosome organization segregation.

## Results

**AcpP interacts with the MukB coiled-coils.** The nature and function of the interaction between AcpP and MukB has been unclear, despite numerous reports describing an interaction[16,23,25,28,31,32]. We, therefore, set out to determine whether the interaction between AcpP and MukB is specific and to identify any interaction site on MukB. Wild type (WT) MukB and three truncated variants were purified and tested for the presence of associated AcpP using SDS-PAGE (Fig. 1b, c). Because previous work had shown that truncated MukB hinge mutants did not co-purify with AcpP[17,33,34], we focused on variants containing the ATPase head and head-proximal regions. AcpP co-purified with MukB[HN] (MukB Head-Neck), consisting of the ATPase head and the first ~30% of head-proximal coiled-coils, but not with MukB[H] (MukB Head), consisting of just the ATPase head domain. Even with the addition of recombinant AcpP, no MukB[H]-AcpP binary complexes were detected (Fig. 1d). Consistent with these observations, AcpP co-purified with MukB[N] (MukB Neck) consisting of just the head-proximal coiled-coils (Fig. 1c). Analysis of samples containing AcpP and MukB[N] or MukB[HN] using native Mass Spectrometry (nMS) revealed AcpP interacts with MukB with a 1:1 monomer-monomer stoichiometry (Fig. 1e, f), supporting data previously reported for WT MukB[16]. In addition, complexes with a mass corresponding to MukB[N]$_2$-AcpP$_2$ were also identified, likely arising through interactions between the coiled-coils. No such dimers were detected in MukB[HN]-AcpP samples, demonstrating that the presence of the ATPase heads inhibits such interactions.

To identify the MukB–AcpP interface, we utilized in vitro chemical cross-linking mass spectrometry (XL-MS). Treatment of MukB[HN] with BS[3] cross-linker in the absence of AcpP, generated a mixture of inter- and intramolecular cross-links (indicated by a top arrow and bottom arrows, respectively; Supplementary Fig. 1a). In the presence of AcpP, despite the lack of detectable MukB[HN]-AcpP cross-links, we noted the disappearance of three substantial species, whose analysis by XL-MS showed that AcpP inhibited the formation of three intramolecular MukB[HN] cross-links involving residue K1125, and one intermolecular cross-link between two K1125 residues of separate MukB[HN] monomers (Supplementary Fig. 1b, 6 and 7). Intermolecular interactions between MukB[HN] molecules are discussed further in subsequent sections. The addition of BS[3] to reconstituted MukBEF-AcpP complexes, identified a cross-link between residue K10 of AcpP and K1125 of MukB, as well as to MukB residues K230 and K1232 (Supplementary Figs. 1b, 6 and 7). Given that K230 and K1232 are in close spatial proximity and K230 is not required to maintain the interaction with AcpP (K230 is not present in MukB[N]), we focused our analysis on residues surrounding K1125. K1125 is located within the C-terminal helix in the coiled-coil

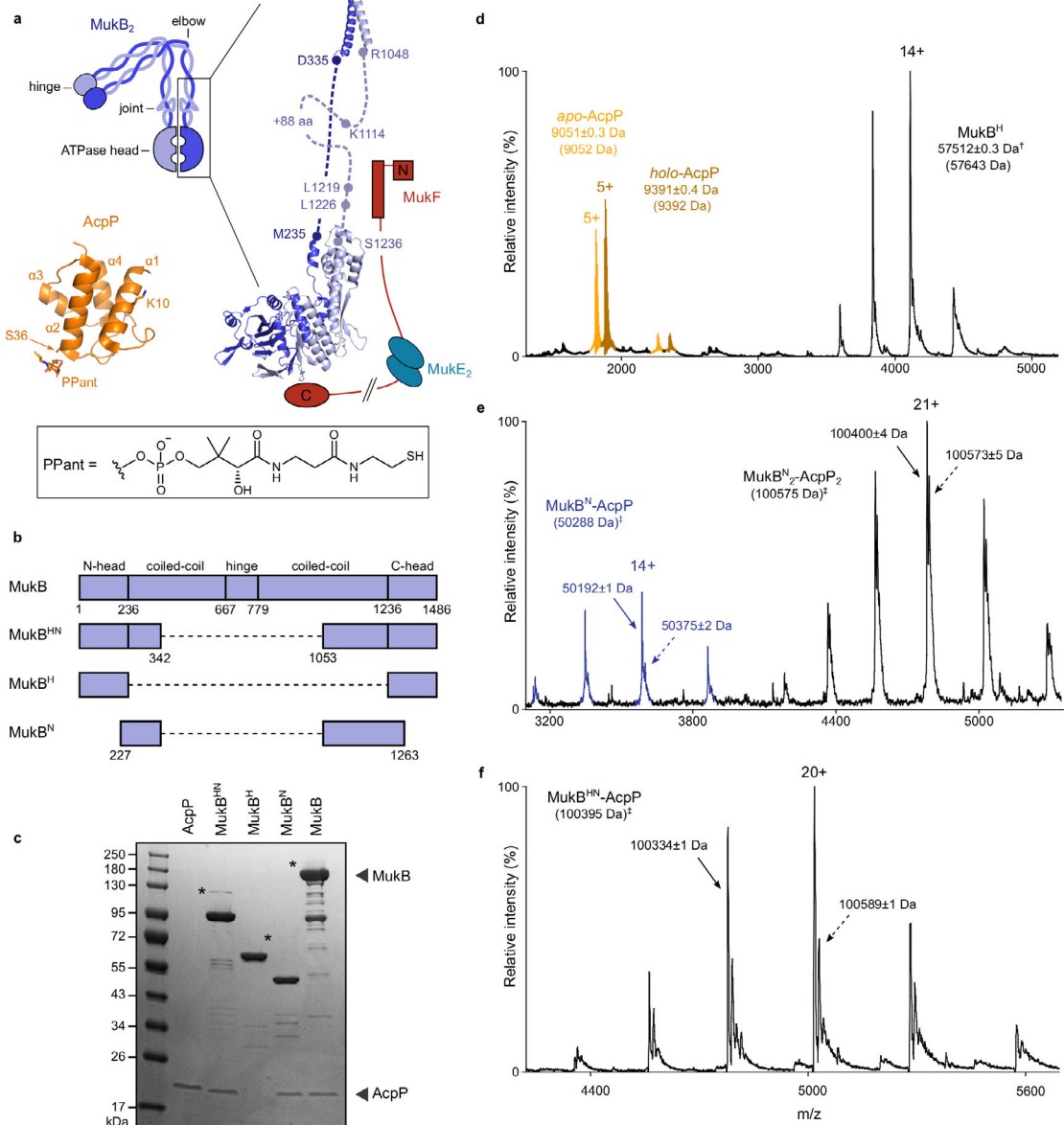

**Fig. 1 Specific binding of AcpP to the neck region of MukB. a** Schematic of MukBEF in the elbow-bent configuration (left); structure of *E. coli* MukB[HN] (right) using a crystal structure of the elbow (PDB 6H2X)[11], and a homology model based on *H. ducreyi* MukB[H] structure (PBD 3EUK)[5]. The coiled-coil and joint are modeled and MukEF are shown in cartoon form (note that the C- and N-terminal domains of a given MukF monomer normally contact different MukB monomers). Structures of AcpP-PPant are also shown (bottom, left, PBD 3NY7)[58]. **b** Schematic of MukB truncations. **c** SDS-PAGE analysis of AcpP co-purification with MukB truncations. Putative disulfide-linked MukB–AcpP species are indicated with an asterisk. Note that AcpP (MW 8640 Da) runs with an apparent MW of ~18,000 Da on SDS-PAGE. The gel is representative of at least two independent experiments. **d–f** nMS analysis of AcpP–MukB truncation interactions. **d** MukB[H] with the addition of recombinant AcpP (mixed population of apo and holo species), †difference of 131 Da suggests N-terminal Met excision. **e** MukB[N] with co-purified AcpP and **f** MukB[HN] with co-purified AcpP. Theoretical masses in parentheses. ‡An averaged mass of apo- and holo-AcpP was used in the theoretical mass calculations. In samples containing native AcpP, at least two different species were identified, likely representing different posttranslational modifications of AcpP. Source data are provided as a Source Data file.

proximal to the ATPase head domain (Fig. 1a, b and Supplementary Fig. 1b). Crystal structures of the MukB elbow and ATPase head indicate the C-terminal helix in this region includes an additional ~80 residues compared to the N-terminal helix and likely forms a conserved joint motif, which was also evident in cross-linking experiments[7–9,13] (Fig. 1a). Sequence alignment of MukB proteins around K1125, indicates high conservation of this and other basic residues, K1114 and R1122 (Supplementary Fig. 1c).

Other characterized AcpP-partner protein interfaces involve electrostatic interactions centered on the six acidic residues in the

α2 helix[35,36] (Fig. 1a; residues 36–50). This also seems to be true for the MukB–AcpP interface, as substitutions in the α2 helix of AcpP abolished its co-purification with MukB[28,32]. The α2 helix contains none of the four lysine residues within AcpP. AcpP residue K10, which is cross-linked to MukB K1125E, is in the α1 helix. Therefore, we reasoned that the three highly conserved basic residues that we identified in MukB might well comprise at least part of the MukB–AcpP interface. Accordingly, we mutated residues K1114–K1125 to glutamic acid in an attempt to perturb the AcpP–MukB interface. In addition, we constructed a double and triple charge reversed MukB mutant, MukB[KK] (containing

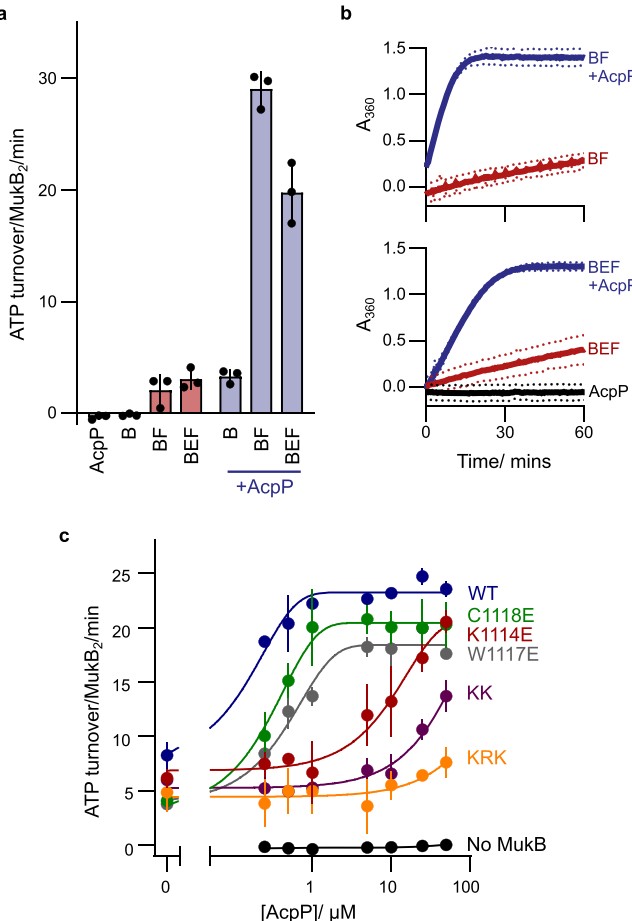

**Fig. 2 MukB ATPase activity requires interaction with AcpP. a** Initial rate ATPase activity measurements of MukB in the presence and absence of AcpP (±SD from three technical repeats). **b** Absorbance data showing the measured activity of MukB over a time course of 60 min (dotted lines represent SD from three technical repeats). **c** Initial rate ATPase activity measurements of MukB proteins in response to increasing concentrations of AcpP (±SD from three technical repeats). MukF and MukE were included at a constant concentration in all samples. Source data are provided as a Source Data file.

the K1114E and K1125E mutations) and MukB$^{KRK}$ (containing K1114E, R1122E, and K1125E mutations). We observed a reduction in the levels of co-purified AcpP in MukB$^{K1114E}$, MukB$^{W1117E}$, and MukB$^{C1118E}$ samples, as judged by SDS-PAGE, confirming the importance of these residues to the MukB–AcpP interface (Supplementary Fig. 2c, d). We also observed a loss of AcpP co-purification in the MukB$^{KK}$ and MukB$^{KRK}$ samples. Together, these results provide strong evidence for a specific AcpP binding site located at the joint, within the MukB coiled-coils.

**AcpP is required for MukB ATPase activity in vitro.** To characterize the functional significance of the MukB–AcpP interaction, AcpP was depleted from WT MukB during heparin purification using an extended salt gradient, where AcpP-depleted MukB eluted as a second peak with a higher retention time (Supplementary Fig. 2a, b). We then sought to identify any effects of removing AcpP on the ATPase activity of MukB. No detectable ATP hydrolysis was observed for the AcpP-depleted MukB sample and only minimal activity was seen as a result of MukB activation by MukF (2.0 ± 1.4 ATP molecules/MukB$_2$/min; Fig. 2a, b). Remarkably, the addition of recombinant AcpP

restored ATPase activity to MukBF complexes (from 3.3 ± 0.6 (−AcpP) to 29.0 ± 1.6 (+AcpP) ATP molecules/MukB$_2$/min), to a level comparable to MukBF co-purified with AcpP (27.2 ± 1.2 ATP molecules/MukB$_2$/min) and similar to that reported previously (where the samples will have contained co-purified AcpP)[14]. Consistent with this, the addition of MukE to AcpP containing MukBF samples modestly inhibited MukF activation (Fig. 2a, b), as reported previously[14]. In these experiments, recombinant AcpP was a mixed population of apo- and holo-AcpP (lacking or containing the PPant prosthetic group, respectively). The relative contributions of these forms are explored later.

To address how AcpP activates MukBEF ATPase, we explored whether AcpP is necessary to assemble a functional MukBEF complex; for example, for the binding of MukF(E) to MukB, given that both AcpP and MukF are required for MukB ATPase. Since MukB residues K1114–K1125 are in proximity to residues L1219 and L1226, which have been implicated in MukF N-terminal domain binding[14] (Fig. 1a), we used nMS and blue native gel electrophoresis (BN-PAGE) to assay the formation of MukBEF complexes in the presence and absence of AcpP. nMS analysis of mixtures of MukB, E, F and AcpP identified complexes consistent with a MukB$_2$E$_4$F$_2$ stoichiometry with one or two AcpP molecules bound. In addition, complexes with masses corresponding to MukB$_4$E$_4$F$_2$ and three or four molecules of bound AcpP were also observed in nMS (Fig. 3a). These MukB dimer of dimer (DoD) complexes, whose abundance increased as the MukB concentration was raised (Fig. 3a), arise when a MukF dimer binds two separate MukB dimers (Fig. 3b). Complementary BN-PAGE experiments with a monomeric MukF derivative[14], confirmed that DoD complexes depend on MukF dimerization (Fig. 3d). Furthermore, the formation of dimer and DoD complexes was independent of AcpP (Fig. 3c), thereby demonstrating that AcpP binding to MukB is not required for the interaction of MukB with MukEF to form either dimeric or DoD complexes. These experiments also show that the formation of MukBEF DoD complexes requires neither bound nucleotide, nor head engagement.

MukB$^{K1114E}$ and MukB$^{KRK}$ mutant proteins, which are deficient in AcpP binding, were also able to form complexes with MukEF (Fig. 3e). Additionally, MukEF formed complexes with MukB$^{HN}$ variant proteins (K1114E and C1118E) (Supplementary Fig. 3d). These data, taken together with the results of in vivo analysis showing the formation of MukF-dependent chromosome-associated MukB$^{KRK}$EF foci (later), demonstrate that the AcpP interaction with MukB is not a prerequisite for MukF binding.

**AcpP binding to the MukB joint inhibits intermolecular interactions.** To gain further insight into how AcpP influences the conformation of MukBEF complexes and how this might lead to productive cycles of ATP hydrolysis, we analyzed MukB$^{HN}$ complexes that form with MukEF in the presence and absence of AcpP and which are more amenable to stoichiometry analysis on native gels. This approach was informed by our initial demonstration that AcpP binding perturbs the formation of a BS$^3$-induced intermolecular cross-link between two K1125 MukB$^{HN}$ residues in the joint (Supplementary Fig. 1a, b). Neither dimeric MukB$^{HN}_2$E$_4$F$_2$ complexes, nor the equivalent of DoD complexes, which form by head engagement in the presence of AMPPNP (MukB$^{HN}_4$E$_4$F$_2$)[14] required AcpP (Supplementary Fig. 3a, c, left-hand panels; b). The addition of recombinant AcpP gave the same complexes but with AcpP bound (Supplementary Fig. 3a, c, right-hand panels; b). Furthermore, in the absence of AcpP, we also observed slower migrating larger intermolecular MukB$^{HN}$EF complexes that we

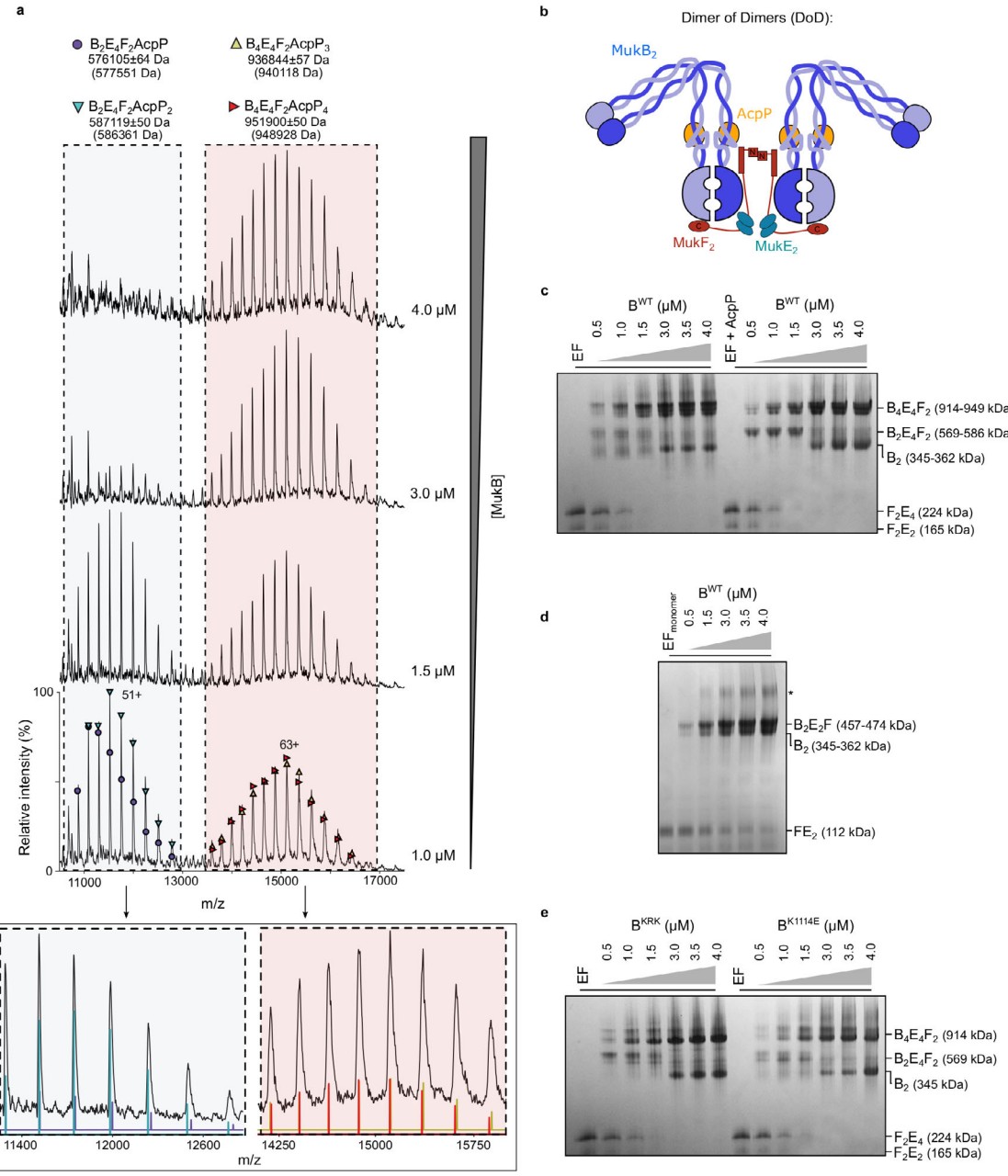

**Fig. 3 MukBEF forms DoD complexes independent of AcpP binding. a** nMS analysis of MukBEF-AcpP complexes at various concentrations of MukB. **b** Schematic of MukBEF DoD complexes, the approximate position of the AcpP binding site is indicated. **c–e** BN-PAGE analysis of complex formation in MukBEF-AcpP showing; (**c**) DoD complex formation is not dependent on the presence of AcpP, (**d**) higher-order, DoD, complexes require the presence of dimeric MukF and (**e**) MukB^KRK and MukB^K1114E still form DoD complexes. Note co-purification of endogenous MukF with recombinant MukB led to the formation of MukB$_4$E$_4$F$_2$ complexes in samples containing monomeric F, indicated with an asterisk. The gels are representative of at least two independent experiments and theoretical masses are shown in parentheses. Source data are provided as a Source Data file.

propose have the stoichiometries MukB$^{HN}_4$E$_8$F$_4$ (−AMPPNP) and MukB$^{HN}_8$E$_8$F$_4$ (+AMPPNP); these disappeared in the presence of AcpP (Supplementary Fig. 3a, c; compare left- and right-hand panels). We propose that the higher-order MukB$^{HN}_8$E$_8$F$_4$ (+AMPPNP) and MukB$^{HN}_4$E$_8$F$_4$ (−AMPPNP) complexes arise from the dimerization of MukB$^{HN}_4$E$_4$F$_2$ and MukB$^{HN}_2$E$_4$F$_2$ complexes, respectively, through coiled-coil interactions in the AcpP binding region of the joint where the K1125 residues were cross-linked by BS$^3$ (Supplementary Fig. 3e).

The single substitution mutant proteins (K1114E and C1118E) failed to produce these presumptive higher-order complexes, irrespective of the presence of AcpP, but still formed AMPPNP-dependent MukB$^{HN}_4$E$_4$F$_2$ complexes, independent of AcpP (Supplementary Fig. 3d). This indicates that the glutamate substitution in these proteins is sufficient to disrupt the intermolecular coiled-coils interaction characterized here. Consistent with AcpP perturbing intermolecular coiled-coil interactions between joint regions, we observed that higher-order bands,

formed through a presumptive disulfide interaction involving residue C1118, were also inhibited by AcpP (Fig. 2c and Supplementary Fig. 2b). Any functional significance of the intermolecular interactions between the coiled-coil joint regions observed here and their inhibition by AcpP remains to be determined, as does understanding whether the inhibition by AcpP is a consequence of a steric constraint, or by AcpP inducing a conformational change in the MukB coiled-coils.

**Mutagenesis of the MukB joint region impairs AcpP-activated ATPase.** To further analyze the requirement of AcpP binding for MukB ATPase activity, we analyzed the mutant proteins that failed to co-purify with AcpP (MukB$^{K1114E}$, MukB$^{W1117E}$, MukB$^{C1118E}$, MukB$^{KK}$ and MukB$^{KRK}$) (Fig. 2c and Supplementary Fig. 2c, d). All five mutant proteins showed low ATPase activity in the presence of MukEF, in contrast to the mutants that co-purified with AcpP, which exhibited levels consistent with the amount of AcpP present within the sample (compare Supplementary Fig. 2d, e). The mutant proteins that lacked co-purified AcpP were then tested to see if the addition of recombinant AcpP stimulated their ATPase activity. AcpP-depleted WT MukB regained maximal ATPase activity after the addition of a 2-fold molar excess of AcpP (Fig. 2c). MukB$^{W1117E}$ and MukB$^{C1118E}$ both regained maximal ATPase activity with a 2–10 fold molar excess of AcpP, suggesting that these substitutions had only a modest impact on the MukB–AcpP interface, despite the conservation of these residues in MukB proteins (Supplementary Fig. 1c). The charge reversal mutants, MukB$^{K1114E}$, MukB$^{KK}$, and MukB$^{KRK}$, showed a sequential reduction in the ability of AcpP to stimulate ATPase activity. At 100 times AcpP excess (50 μM; comparable to the in vivo cellular concentration) the activity of MukB$^{KRK}$ was only $7.6 \pm 1.4$ ATP molecules/MukB$_2$/min (~32% of the WT MukBEF activity in the presence of 50 μM AcpP) (Fig. 2c). These data support the conclusion that AcpP binding to MukB promotes in vitro ATPase activity.

**MukB ATPase activity is stimulated by both apo- and holo-AcpP.** AcpP overexpression in E. coli results in a mixture of both apo-AcpP and holo-AcpP species (Supplementary Fig. 4a). These species can be interconverted after purification with the use of a recombinant phosphodiesterase (AcpH), which removes the PPant modification leaving apo-AcpP, or phosphopantetheinyl transferase (Sfp), which covalently adds the PPant group to residue S36[31,37,38]. Further modification of the PPant group through the covalent interaction of acyl groups within the cell, generates a plethora of acylated AcpP intermediates[39]. We, therefore, investigated whether posttranslational modification of AcpP is required for its interaction with MukB. Analysis of MukB by nMS indicated the presence of two AcpP species within the sample, likely apo- and holo-AcpP (Fig. 1e, f). Furthermore, we commonly observed additional bands on SDS-PAGE, sensitive to reducing agent, that ran with a higher molecular mass than purified MukB, or its truncated variants, MukB$^{HN}$ and MukB$^{N}$ (indicated with an asterisk in Figs. 1c and 2c). Analysis of these bands with anti-AcpP antibody and proteomic MS demonstrated the presence of AcpP (Supplementary Fig. 4b). These bands were also observed in a selection of MukB neck mutants including MukB$^{G1116E}$, MukB$^{W1117E}$ and MukB$^{V1124E}$, but absent in the MukB$^{C1118E}$ sample, suggesting the formation of a disulfide bond between C1118 and the free thiol of holo-AcpP (Supplementary Fig. 4b). This disulfide interaction was unnecessary for in vitro ATPase stimulation, as both apo- and holo-AcpP could stimulate MukB ATPase to the same extent (Supplementary Fig. 4c). In addition, cells expressing MukB$^{C1118E}$ were viable and displayed apparent WT MukBEF activity (see below). Nevertheless, the

formation of this disulfide bond could contribute to the stabilization of the AcpP–MukB interaction.

**AcpP-deficient MukBEF complexes have perturbed in vivo behavior.** Next, we assessed the viability of MukB mutants impaired in AcpP binding by transforming plasmid-borne genes of the mutants into a ΔmukB background strain. ΔmukB cells exhibited temperature-sensitive growth in the rich medium at 37 °C, which was restored by basal expression from the multi-copy number plasmid pET21a encoding a WT mukB gene (Supplementary Fig. 5a). All single and double MukB mutants, which were deficient in AcpP binding in vitro, had a Muk$^+$ phenotype, as assessed by growth at 37 °C. In contrast, cells expressing MukB$^{KRK}$ exhibited temperature-sensitive growth at 37 °C, consistent with the ATPase activity defect in this mutant and particularly the substantially impaired response to added AcpP; the ~32% residual ATPase activity in the presence of 50 μM AcpP is noteworthy given that this is close to the estimated in vivo AcpP concentration (Fig. 2c). MukB$^{KR}$ (K1114E, R1122E), MukB$^{RK}$ (R1122E, K1125E) and MukB$^{KC}$ (K1114E, C1118E) cells were Muk$^+$, as assessed by growth at 37 °C, indicating that the temperature sensitivity of MukB$^{KRK}$ is likely due to a lack of AcpP interaction, although we cannot eliminate the possibility that the phenotype is influenced by protein conformational changes induced by the combined mutations. Consistent with our observations, multiple substitutions in other AcpP-target protein interfaces are required to abolish AcpP binding with other AcpP binding proteins in addition to MukB[40,41].

We then explored the functional consequences of the impaired MukB–AcpP interactions by analyzing the behavior of WT and mutant MukBEF complexes by quantitative live-cell imaging. We expressed basal levels of MukB and its variants from the multi-copy number plasmid pBAD24 in ΔmukB cells containing a functional mYpet fusion to the endogenous MukE protein and fluorescent markers located near oriC (ori1) and close to the middle of ter (ter3)[17]. In cells expressing WT MukB, fluorescent MukBEF foci were associated with the ori1 locus, as reported previously by ourselves and others, for MukBEF expressed from the endogenous chromosomal locus (Fig. 4a, b; $57.1 \pm 0.2\%$ colocalization; distances within the diffraction limit (~264 nm))[6,17–19]. Consistent with this, only $7.9 \pm 0.3\%$ of MukBEF foci colocalized with ter3. In contrast, MukB$^{EQ}$EF foci colocalized with ter3 and not ori1, as reported previously, because they remain associated with MatP bound to matS sites within ter, as a consequence of their defect in ATP hydrolysis[6,17,18]. A MukB mutant that does not bind ATP (MukB$^{DA}$), had its MukBEF distributed over the whole nucleoid, with few, if any, defined fluorescent foci (Fig. 4a)[6,17,18].

The AcpP binding-impaired variants of MukB all produced fluorescent MukBEF foci, thereby demonstrating the association of their clustered MukBEF complexes with the chromosome. The mutants fell into two classes; those indistinguishable from the pattern of WT MukB focus distribution (MukB$^{W1117E}$, MukB$^{C1118E}$, and MukB$^{K1125E}$) and those that had a reduced ori1 association and increased ter3 association. These latter variants all contained the MukB$^{K1114E}$ mutation either alone, or in combination with one or two further mutations in the AcpP binding region, MukB$^{KK}$ and MukB$^{KRK}$, respectively. MukB$^{K1114E}$, showed a small reduction in association with ori1 ($47.7 \pm 2.0\%$) and a complementary increase in association with ter3 ($14 \pm 1.3\%$) (Fig. 4b). MukB$^{KK}$ and MukB$^{KRK}$ shared almost identical MukBEF focus properties; $35.7 \pm 1.2\%$ and $35.8 \pm 1.2\%$ colocalization with ori1, respectively, and substantially increased association with ter3 ($25.2 \pm 0.9\%$ and $21.0 \pm 0.3\%$ ter3 colocalization, respectively). Despite these similarities, only MukB$^{KRK}$ cells exhibited temperature-sensitive growth, while the double mutants, like the single ones, grew at 37 °C. The behavior of the

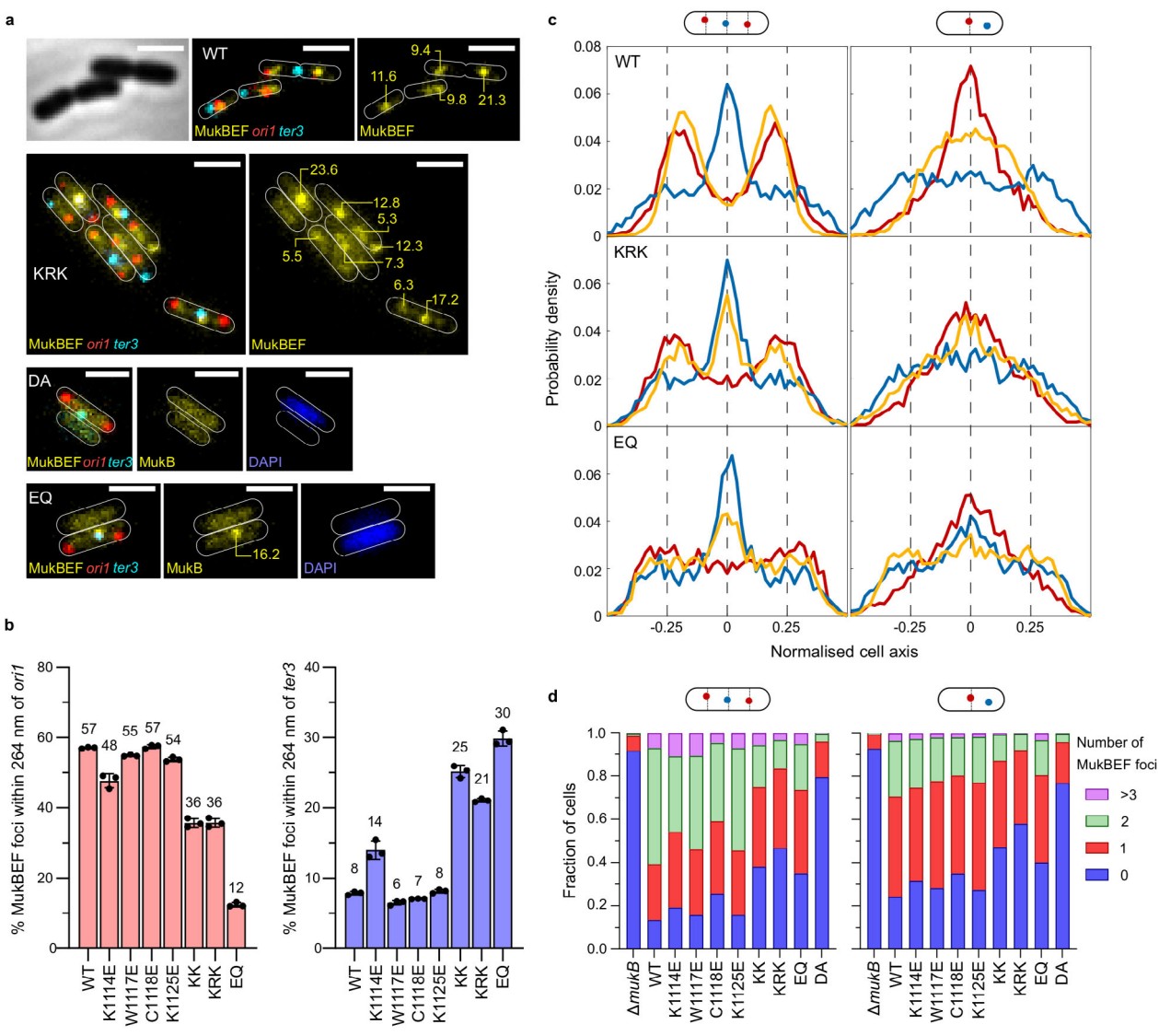

**Fig. 4 MukBEF complexes that are deficient in AcpP binding have perturbed behavior in vivo.** *ΔmukB* cells with fluorescently labeled MukE (mYPet), *ori1*(mCherry), and *ter3* (mCerulean) were grown in minimal glycerol medium at 30 °C. Under these conditions, basal expression from a pBAD24 plasmid encoding WT MukB was sufficient to confer a Muk+ phenotype on cells. **a** Representative images of the indicated strains used in subsequent analysis. The numbers on the images indicate relative brightness of the foci. Scale bars: 2 μm. **b** Colocalization of fluorescent MukBEF complexes with *ori1* and *ter3* for the indicated cells (MukB$^{WT}$ 8534 cells, MukB$^{K1114E}$ 7862 cells, MukB$^{W1117E}$ 5402 cells, MukB$^{C1118E}$ 9446 cells, MukB$^{K1125E}$ 9911 cells, MukB$^{KK}$ 5900 cells, MukB$^{KRK}$ 3676 cells, and MukB$^{EQ}$ 3849 cells; ± SD from three biological repeats). **c** Position of MukBEF foci relative to *ori1* and *ter3*, with respect to the cell axis for all analyzed cells. **d** Histograms showing the number of fluorescent MukBEF foci/cell with respect to *ori1* and *ter3*. Left panel, cells with 2 *ori1* loci and 1 *ter3* locus. Right panel, cells with a single *ori1* focus because the locus has not replicated/segregated. Source data are provided as a Source Data file.

mutants in relation to *ori1/ter3* localization was independent of whether there was a single *ori1* locus present (in cells soon after birth that had not replicated or segregated the *ori1* locus), or whether there were two sister *ori1* loci, after replication and segregation (Fig. 4c). Nevertheless, we noted that the double and triple mutant containing cultures had an increasing proportion of cells with no detectable fluorescent MukBEF foci (38 ± 2% and 43 ± 1%, respectively),

compared to only 12 ± 2% in WT MukB cells (Fig. 4d), suggesting that a significant proportion of cells had most if not all of their mutant MukBEF complexes defective in ATP binding and chromosome association.

The progressive shift from *ori1* to *ter3* colocalization in mutants carrying the MukB$^{K1114E}$ mutation was further evident when the normalized distribution of *ori1*, *ter3* and MukBEF foci

along the longitudinal cell axis was plotted (Fig. 4c). In cells expressing MukB[KRK] that had 2 ori1 loci at ¼ and ¾ positions on the long cell axis, a large proportion of MukBEF foci were at the cell center where ter3 is preferentially located, in addition to the ¼ and ¾ positions. This phenotype is intermediate between cells expressing WT MukB and those expressing MukB[EQ], which binds ATP but is deficient in hydrolysis (Fig. 4a–c)[17]. The intermediate MukB[KRK] phenotype was also reflected in a slight shift in ori1 positioning from the ¼ and ¾ positions towards the poles, which was more evident in MukB[EQ] cells, as well as in cells lacking MukB (Fig. 4c)[19].

We next analyzed the percentage of anucleate cells within each population, assessed by lack of DAPI staining and absence of ori1 and ter3 fluorescent markers (Supplementary Fig. 5b). Cell populations lacking mukB or expressing MukB[EQ] or MukB[DA] contained $12.4 \pm 2.4$, $9.0 \pm 3.0$ and $6.9 \pm 0.6\%$ anucleate cells, respectively, whereas WT populations contained $1.0 \pm 0.3\%$ anucleate cells. These observations are consistent with the temperature-sensitive growth of the MukBEF-impaired cells and either lack of MukBEF foci (ΔmukB, mukB[DA]), or abnormally positioned foci (mukB[EQ]). MukB[KRK] expressing cells contained $2.4 \pm 2.0\%$ anucleate cells, likely higher than WT cells, but less than the Muk⁻ cells, whereas MukB[K1114E] and MukB[KK] expressing cells had close to WT levels of anucleate cells ($1.4 \pm 0.2\%$ and $1.1 \pm 0.1\%$, respectively). Overall, these observations are consistent with an increasing impairment in MukBEF function as AcpP binding is decreased in a stepwise fashion by a progressive mutation in the AcpP binding region of MukB. Despite MukB[KRK] cells still being able to form chromosome-associated MukBEF complexes, we propose that at least a substantial fraction of these are impaired in MukBEF function, consistent with a defect in ATP hydrolysis, temperature-sensitive growth, increased anucleate cell formation, and consequent preferential location within ter.

Cells expressing MukB[G1116E] and MukB[V1124E] also exhibited temperature sensitivity, although the defect was not as complete as for ΔmukB cells. <10% of MukB[V1124E] plated cells yielded colonies at 37 °C, with the surviving colonies being relatively small. A higher proportion of MukB[G1116E] expressing cells grew at 37 °C, but the colonies were again smaller (Supplementary Fig. 5a). The basis for this sensitivity in MukB[G1116E] cells is not clear, as cells grown at 30 °C in minimal media had a WT MukB⁺ phenotype as assessed by fluorescent MukBEF foci that are ori1-associated and not ter3-associated (Supplementary Fig. 5c, d). In contrast, cells expressing MukB[V1124E] displayed no clear MukBEF foci, but diffuse mYPet fluorescence similar to cells containing MukB[DA] (Supplementary Fig. 5b). Despite interacting with AcpP and demonstrating moderate ATPase activity in vitro, MukB[V1124E] seemed unable to interact stably with the chromosome, presumably because the mutation directly interferes with MukBEF function (Supplementary Fig. 5c). The observation that mutations in this region of the MukB coiled-coil can interfere with AcpP binding, or otherwise influence MukB function, underlines the functional importance of the joint region in SMC complexes.

## Discussion

We have characterized the specific interaction of AcpP with the joint region of the MukB coiled-coils and have shown that it is necessary for MukB ATPase activity in vitro and for normal MukBEF function in vivo. A recent CryoEM study has confirmed the AcpP–MukB interaction revealed here, but provided no insight into the functional significance[42]. The cellular consequences of the MukB–AcpP interaction remain to be determined; in particular, understanding whether AcpP binding to MukBEF in vivo is constitutive and unregulated, or whether it is modulated during cycles of MukBEF action, and/or by cellular metabolism. Activation of MukB ATPase activity by AcpP binding, underlines the importance of the joint whose functional roles are only now being revealed. This is emphasized by our demonstration that other mutations in the AcpP binding region of the MukB joint, which do not affect AcpP binding, can perturb MukB function, whether it be impaired ATPase, or in vivo action.

The molecular mechanism by which AcpP activates MukB ATPase activity and overall MukBEF action remains unknown. The AcpP binding site at the MukB joint is relatively distant from the ATPase head and the "bent elbow" configuration of MukB occurs in the absence of bound AcpP[11]. The SMC joint is highly conserved[7,8] and can be bound by other SMC accessory proteins[43]. Studies of both prokaryote and eukaryote SMC complexes have led to proposals that conformational flexibility in the coiled-coils, facilitated by the plasticity of the joint, allows transitions in the disposition of the two SMC heads during their juxtaposition, engagement, and disengagement during cycles of ATP binding and hydrolysis. These must be coupled with changes in DNA association during presumed loop extrusion by the complexes[7,8]. We favor the view that AcpP binding to the MukB joint modulates such transitions. Since AcpP is acidic and the MukB region involved in its interaction is basic (Supplementary Fig. 1c), it is possible that DNA and AcpP, compete at least transiently, for association with the joint region during these transitions. We have shown that AcpP binding to the MukB joint is not required for MukBEF complex assembly, nor is it required for nucleotide- and MukEF-dependent head engagement in the truncated MukB[HN] variant, as assessed by native gel electrophoresis. Nevertheless, as the disposition of MukB[HN] ATPase heads are not constrained by the elbow, hinge, or the rest of the coiled-coils, the MukB[HN] head engagement that we assay may not reflect the conformational changes that are likely necessary during head juxtaposition and engagement of the full-length protein[2,7,44,45].

Our observation here that dimer of dimer (DoD) complexes of full-length MukB complexed with MukEF, the functional unit in vivo[6], can be detected in vitro in the absence of bound AcpP, or AMPPNP-induced head engagement (Fig. 3 and Supplementary Fig. 3), demonstrates that the configuration of two ATPase heads of a full-length MukB dimer prevents two MukF C-terminal domains of a MukF dimer binding to the same MukB dimer, even in the absence of head engagement. Our favored interpretation is that the proximity of the hinge to one of the heads, in the elbow-bent configuration (Fig. 1a), generates an asymmetry, in a way similar to that induced by head engagement[5], so that only one MukF C-terminal domain can bind a head in a MukB dimer; leaving the other C-terminal domain to capture a second MukB dimer (Fig. 3b). An alternative model in which the disposition of unengaged heads is constrained by relatively rigid coiled-coils in the neck region, again allowing only one MukF C terminus to bind a MukB dimer, seems less likely.

Given that other SMC complexes can act in the absence of AcpP binding to the joint, it is difficult to rationalize why this requirement has evolved in the MukBEF clade; there is no obvious connection between AcpP and the other MukBEF co-evolved players that include MatP, SeqA, Dam, and topoisomerase IV[17,46]. AcpP is highly abundant in E. coli cells[26] (~100 μM; a $>10^2$-fold cellular molar excess over endogenous MukBEF) and is involved in a wide range of essential steps in fatty acid biosynthesis, along with other specific interactions. Since it exists in a wide range of acylated and unacylated forms, it is challenging to imagine how any modulated MukBEF activity on chromosomes results from cellular changes in AcpP as a consequence of changes in fatty acid metabolism. Parenthetically, MukBEF function only

becomes essential for cell viability under condition of rapid growth during which overlapping rounds of replication occur[18]. Indeed, the MukBEF clade of SMC complexes is largely confined to bacteria that support overlapping rounds of replication as part of their lifestyle. Nevertheless, MukBEF is clearly active and important for normal chromosome organization-segregation under conditions of slow growth, when each round of replication is initiated and terminated in the same cell cycle[6,17,18]. Although our work has not identified any specific form of AcpP that preferentially interacts with MukB or influences its activity, any connection between cellular metabolism and the activity of MukBEF complexes on the chromosome, is likely to involve a specific form (or forms) of AcpP whose abundance and activity is under metabolic control. In this scenario, levels of fatty acid biosynthesis could be coordinated in some way with chromosome organization segregation mediated by MukBEF. Our assays have found no evidence for this; apo-AcpP and holo-AcpP had comparable activities in stimulating MukB ATPase in vitro, while a disulfide between the PPant free thiol and MukB[C1118] is not essential for either ATPase or in vivo function. An alternative scenario to one in which the AcpP–MukB interaction modulates MukBEF action with fatty acid and lipid synthesis is one in which this is an "accidental" recruitment of a protein during evolution, just like the recruitment of the "metabolic enzymes", ArgR, ArcA and PepA, as essential accessory factors in site-specific recombination essential for multi-copy plasmid stability[47,48].

Elsewhere, it has been proposed that the interaction of AcpP with proteins uninvolved in acyl transfer may contribute to the coordination of cellular metabolism. For example, the SpoT–AcpP interaction may help coordinate the cells protein synthesis stringent response to fatty acid starvation[29,32]. Similarly, the interaction between AcpP and the SecA component of the protein-membrane translocase machinery could couple fatty acid–lipid metabolism with protein transport through the inner membrane. Although it has been proposed that binding of AcpP to MukB might mediate interactions with the SecA component of the protein-membrane translocase machinery, to allow for correct oriC positioning within cells[49,50], in our opinion this appears unlikely. A Turing patterning mechanism positions the largest cluster of MukBEF complexes on the chromosome at either midcell or ¼ positions and the ori association with these clusters results directly from the depletion of MukBEF complexes from ter as a consequence of their dissociation directed by their interaction with MatP-matS[18]. We are unaware of any compelling evidence that replication origins are associated either with SecA complexes or the inner membrane.

The perturbed ori1 positioning in AcpP binding defective MukB[KRK] expressing cells is similar to that observed in other situations where MukBEF function is impaired sufficiently to give a temperature-sensitive growth phenotype, regardless of whether it is a defect in ATP binding (MukB[DA]), hydrolysis (MukB[EQ]), or where there is a complete lack of MukB. The ability of MukB[KRK] expressing cells to form fluorescent clusters of MukBEF complexes demonstrates that under conditions of impaired AcpP binding, these complexes can still associate with the chromosome, despite a substantial fraction of these being impaired in MukBEF function. These observations are consistent with a defect in ATP hydrolysis arising from impaired AcpP binding, with the consequent preferential location of chromosomal-bound complexes to within ter, similar to ATP hydrolysis-defective MukB[EQ]EF complexes that cannot be displaced from MatP-bound matS sites within ter, because of impaired ATP hydrolysis[6,17,18]. Since a proportion of cellular MukB[KRK] is likely to be bound by AcpP, given the latter's abundance, we believe this explains why some MukB[KRK] complexes are ori-associated and at least partly functional, albeit with cells having a Muk⁻ phenotype as assessed by temperature sensitivity. In a situation where MukB could not bind AcpP at all, we do not know whether the disposition of the heads would allow sufficient ATP binding to associate with ter as in mukB[EQ] cells, or whether ATP binding would be so transient that few if any chromosome-associated complexes would be present, as in mukB[DA] cells. The demonstration of a progressive defect in the ability of MukB to respond to increasing AcpP in ATPase assays when the MukB[K1114E] substitution is combined with one (MukB[KK]), or two further substitutions (MukB[KRK]) in the AcpP binding region of MukB, correlates well with the increasing in vivo defects that indicate impaired ATPase hydrolysis. Also, the MukB[KK] mutant, which had 60% of WT ATPase activity in vitro at 50 μM AcpP has a Muk⁺ phenotype in vivo, while MukB[KRK], with 32% ATPase activity in vitro at 50 μM AcpP was Muk⁻. By comparison, a study of the relationship between the in vitro ATPase activity of B. subtills mutant SMC complexes and their rate of DNA translocation in vivo, which can be assessed directly in this organism, showed mutants that had 6-20% residual ATPase activity in vitro retained 60–80% of DNA translocation activity in vivo and had an almost normal Smc⁺ phenotype assessed by growth[51].

The work reported here, provides the platform for future studies of the MukBEF mechanism and how it is influenced by AcpP. This will require an integrated combination of structural, biochemical, biophysical, and genetic studies and may elucidate more mechanistic and functional insights into the MukBEF clade of proteins, which has evolved an apparently unique architecture, along with a distinctive family of co-evolved partners.

## Methods

**Protein overexpression and purification**. MukB-His, MukE-His and His-MukF were overexpressed from pET vectors in C3013I cells (NEB), MukB-His variants were expressed in strain FL01, which is mukB 3xFLAG C3013I (NEB)[14]. Proteins were purified as previously described[14]. Briefly, after purification using TALON superflow resin, protein samples were further purified using either a HiTrap DEAE FF column (GE Healthcare) for MukE and MukF, or HiTrap Heparin HP column (GE Healthcare) for MukB and derivatives. Appropriate fractions (selected by 4–20% gradient SDS-PAGE) were pooled and concentrated by centrifugal filtration (Vivaspin 20, 5000 MWCO PES, Sartorius) for loading onto a Superdex 200 Increase 10/300 GL (GE Healthcare) column equilibrated in storage buffer (50 mM HEPES pH 7.3, 300 mM NaCl, 1 mM EDTA, 1 mM DTT and 10% (v/v) glycerol). Peak fractions were assessed for purity (>90%) by SDS-PAGE/Coomassie staining, snap-frozen as aliquots, and stored at −80 °C.

AcpP was expressed from a pET28a plasmid encoding acpP with a thrombin-cleavable N-terminal 6xHis tag in C3031I cells (NEB). 2 L cultures of LB supplemented with kanamycin (25 μg/mL) were grown at 37 °C to an $OD_{600}$ of 0.5–0.6 and induced with β-D-1-thiogalactopyranoside (IPTG) at a final concentration of 1 mM. After overnight incubation at 18 °C, cells were harvested by centrifugation, resuspended in lysis buffer (25 mM HEPES, 150 mM NaCl, 1 mM TCEP, 10% glycerol) supplemented with a protease inhibitor tablet and homogenized. Cell debris was removed by centrifugation and cell lysate mixed with ~5 mL of TALON Superflow resin and incubated for 30 mins at 4 °C. The slurry was poured into a column and washed with 10× volume lysis buffer, 4× volume wash buffer A (25 mM HEPES, 150 mM NaCl, 1 mM TCEP, 10% glycerol, 25 mM imidazole), and 1× volume wash buffer B (25 mM HEPES, 150 mM NaCl, 1 mM TCEP, 10% glycerol, 100 mM imidazole). Bound proteins were eluted using elution buffer (25 mM HEPES, 150 mM NaCl, 1 mM TCEP, 10% glycerol, 250 mM imidazole) and dialyzed overnight in lysis buffer with the addition of thrombin protease (10 U per 1 mg of AcpP). Uncleaved protein was removed by incubation with TALON Superflow resin before concentrating for loading onto a Superdex 75 Increase 10/300 GL (GE Healthcare) column equilibrated in lysis buffer. Peak fractions were assessed for purity (>90%) by SDS-PAGE/Coomassie staining, snap-frozen as aliquots, and stored at −80 °C. P. aeruginosa AcpH and B. subtilis Sfp were purified as described[37]. Briefly, 6×His-tagged proteins were first purified using TALON superflow resin. After elution with imidazole, AcpH was immediately desalted using a PD-10 column (Sephadex G-25) into storage buffer (50 mM Tris, pH 8, 100 mM NaCl, 0.5 mM TCEP, 15 mM MgCl₂, 1 mM MnCl₂ and 10% glycerol). Sfp was instead dialyzed overnight into storage buffer (50 mM Tris, pH 8, 10 mM MgCl₂, 0.5 mM TCEP, 10% glycerol). Protein samples were then assessed for purity before aliquoting and storing at −80 °C.

**Maturation of AcpP**. The removal of the AcpP-PPant group was achieved by mixing purified AcpP (1 mg/mL) with AcpH (0.1 mg/mL) in the presence of MgCl₂ (12.5 mM),

MnCl$_2$ (5 mM) and TCEP (5 mM). Reactions were incubated at 37 °C for 24 h and monitored by 20% Urea-PAGE[37]. The addition of the PPant group was achieved in a similar manner, except the final reaction contained 50 mM Tris, pH 7.4, 150 mM NaCl, 10% glycerol, 0.5 mM TCEP, 1 mM CoA, and 0.1 mg/mL *Bs*Sfp. Protein samples were then purified by size exclusion chromatography, snap-frozen as aliquots, and stored at −80 °C.

**ATP hydrolysis assays**. ATP hydrolysis was analyzed in steady-state reactions using an ENZCheck Phosphate Assay Kit (Life Technologies)[14]. 150 µL samples containing standard reaction buffer supplemented with ATP to a final concentration of 1.3 mM were assayed in a BMG Labtech PherAstar FS plate reader at 25 °C. The data were analyzed using MARS data analysis software. Quantitation of phosphate release was determined using the extinction coefficient of 11,200 M–1 cm$^{-1}$ for the phosphate-dependent reaction at A360 nm at pH 7.0. Final reactions contained 65 mM NaCl. Data were then plotted and analyzed in Prism version 8.3.0.

**Native-state ESI-MS spectrometry**. Prior to MS analysis, protein samples were buffer exchanged into 200 mM ammonium acetate pH 8.0, using a Biospin-6 (BioRad) column and introduced directly into the mass spectrometer using gold-coated capillary needles (prepared in-house;). Data were collected on a Q-Exactive UHMR mass spectrometer (ThermoFisher). The instrument parameters were as follows: capillary voltage 1.1 kV, quadrupole selection from 1000 to 20,000 *m/z* range, S-lens RF 100%, collisional activation in the HCD cell 50–200 V, trapping gas pressure setting kept at 7.5, temperature 100-200 °C, resolution of the instrument 12500. The noise level was set at 3 rather than the default value of 4.64. No in-source dissociation was applied. Data were analyzed using Xcalibur 4.2 (Thermo Scientific) and UniDec[52]. Data collection for all spectra was repeated at least three times. Errors on observed masses were calculated by following a previously described method[53].

**Blue-Native gel electrophoresis (BN-PAGE)**. MukB or MukB$^{HN}$ (0–4.5 µM) was incubated with MukF (1.5 µM), MukE (3 µM) and AcpP (at the indicated concentrations) in 4× Native PAGE sample buffer (ThermoFisher Scientific, BN2003) with DTT (1mM) and MgCl$_2$ (1 mM) for 30 min at 22 ± 1 °C. Samples were then analyzed using 3–12% native Bis-Tris gels with dark blue cathode buffer. Gels were destained in 40% (v/v) ethanol, 10% (v/v) acetic acid for 30 min before destaining with 8% (v/v) acetic acid overnight.

**Western blot analysis**. MukB samples were heated to 95 °C in LDS Sample Buffer (4X) (ThermoFisher NP0007) with or without the presence of a reducing agent. Samples were then analyzed using NuPAGE™ 7%, Tris-Acetate SDS-PAGE (ThermoFisher EA03585BOX) followed by western blots using anti-AcpP (LSBio, LS-C370023, 1:5000 dilution) as primary and goat anti-rabbit HPR (ThermoFisher, 65-6120, 1:10,000 dilution) as a secondary antibody.

**Proteomics**. For cross-links involving MukB$^{HN}$, BS$^3$ (50–250× molar excess over MukB$^{HN}$) was added to a sample of MukB$^{HN}$, co-purified with or without AcpP, or with the addition of recombinant AcpP (at various molar ratios). Reactions were incubated at RT for 30 min then quenched with Tris buffer (50 mM) before diluting with SDS-loading buffer and analyzed using SDS-PAGE. Gel bands corresponding to cross-linked species were excised, reduced with TCEP (10 mM), and alkylated with 2-Chloroacetamide (50 mM) before overnight digestion with Trypsin. Peptides samples were speed-vac dried and resuspended in 5% formic acid/5% DMSO before LC–MS/MS analysis. For cross-links involving WT MukB, BS$^3$ (1 mM) was added to samples of AcpP-depleted MukBEF (reconstituted from individually purified proteins) in the presence and absence of recombinant AcpP and AMPPNP (1 mM). Samples were allowed to react for 2 h at RT before quenching with ammonium bicarbonate (100 mM). Samples were then denatured with urea (4 M) in ammonium bicarbonate buffer (100 mM) before the addition of TCEP (10 mM) followed by 2-chloroacetamide (50 mM). Samples were then pre-digested with LysC (1 µg/100 µg of the sample) for 2 h before overnight digestion with trypsin (1 µg/40 µg of sample). Tryptic digestion was stopped with the addition of formic acid (5%). Digested peptides were centrifuged at top speed for 30 min at 4 °C to remove undigested material. The supernatant was loaded onto a handmade C18 stage tip, pre-activated with 100% acetonitrile. Peptides were washed twice in TFA 0.1%, eluted in 50% acetonitrile/0.1% TFA and speed-vacuum dried. Peptides were resuspended into 2% acetonitrile/0.1% formic acid before LC–MS/MS analysis.

Peptides were separated by nano-liquid chromatography (Thermo Scientific Easy-nLC 1000) coupled in line a Q-Exactive mass spectrometer equipped with an Easy-Spray source (ThermoFisher Scientific). Peptides were trapped onto a C18 PepMac100 precolumn (300 µm i.d. × 5 mm, 100 Å, ThermoFisher Scientific) using Solvent A (0.1% Formic acid, HPLC grade water). The peptides were further separated onto an Easy-Spray RSLC C18 column (75um i.d., 50 cm length, ThermoFisher Scientific) using a 120 min linear-gradient (15–35% solvent B (0.1% formic acid in acetonitrile)) at a flow rate of 200 nL/min. The raw data were acquired on the mass spectrometer in a data-dependent acquisition mode (DDA). Full-scan MS spectra were acquired in the Orbitrap (Scan range 350–1500 *m/z*, resolution 70,000; AGC target, 3e6, maximum injection time, 50 ms). The 10 most intense peaks were selected for higher-energy collision dissociation (HCD) fragmentation at 30% of normalized collision energy. HCD spectra were acquired in the Orbitrap at resolution 17,500, AGC target 5e4,

maximum injection time 120 ms with fixed mass at 180 *m/z*. Charge exclusion was selected for unassigned and 1+ ions. The dynamic exclusion was set to 40 s. Tandem mass spectra were searched using pLink software version 2.3.9.[54] against an *E. coli* protein sequence database. Peptide mass tolerance was set at 20ppm on the precursor and fragment ions. Data were filtered at FDR below 5% at the PSM level. Tandem mass spectra of cross-linked peptides were extracted and annotated using pLabel 2.4[55].

**Functional analysis in vivo**. The ability of MukB variants to complement the temperature-sensitive growth defect of a Δ*mukB* strain was tested as described previously, using basal levels of MukB expression from plasmid pBAD24[14]. Live-cell imaging used cells grown in M9 minimal medium with 0.2% (v/v) glycerol, 2 µg/mL thiamine, and required amino acids (threonine, leucine, proline, histidine, and arginine; 0.1 mg/mL) at 30 °C. An overnight culture was diluted ~1000-fold and grown to A$_{600}$ 0.05–0.2 and deposited on a medium containing agarose pad after staining with 1 µg/mL DAPI. The Δ*mukB* cells used had a functional mYpet fusion to the endogenous *mukE* gene, fluorescently labeled *ori1* (mCherry), and *ter3* (mCerulean) (AU2118; *lacO240 @ori1* (3908) (*hyg*), *tetO240/ter3* (1644) (*gen*), Δ*leuB::Plac-lacI-mCherry-frt*, Δ*galK::Plac-tetR-mCerulean-frt*, Δ*araBAD* (AraC+), *mukE-mYPet-T1-T2-Para-ΔmukB-kan*)[17,18], expressing basal levels of pBAD24 plasmid-borne WT MukB, the indicated MukB mutants, or empty pBAD24 plasmid control (Δ*mukB*). Epifluorescence images were acquired on a Nikon Ti-E inverted microscope equipped with a perfect focus system, a ×100 NA 1.4 oil immersion objective (Nikon), an sCMOS camera (Hamamatsu Flash 4), a motorized stage (Nikon), an LED excitation source (Lumencor SpectraX) and a temperature chamber (Okolabs). Fluorescence images were collected with 100 ms exposure time using excitation from a LED source. Phase-contrast images were collected for cell segmentation. Images were acquired using NIS-Elements software (Nikon). Cell segmentation and spot detection from the fluorescence channel were performed using SuperSegger[56]. Low-quality spots were filtered out with a fixed threshold for all data sets (4.5). The threshold was selected to minimize the number of falsely identified MukBEF foci within background signal yet maximize the number of foci analyzed; the threshold ensured ~90% of cells expressing WT MukB contained at least one MukBEF focus, whilst ~90% of Δ*mukB* cells had none. The percentages of cells containing one or more spots, distances to the closest *ori1/ter3* marker, and localization along the long cell axis were calculated using MATLAB (MathWorks) as described[18]. Percentages of anucleate cell formation were calculated using a custom script. Custom scripts used in this study are available in Supplementary Software 1.

**Reporting summary**. Further information on research design is available in the Nature Research Reporting Summary linked to this article.

## Data availability
The data that support the findings of this study are available from the corresponding author upon reasonable request. The mass spectrometry proteomics data generated in this study have been deposited in the ProteomeXchange Consortium via the PRIDE[57] partner repository database with the data set identifiers PXD026017 and PXD026062. Source data are provided with this paper.

## Code availability
The custom scripts used for the quantitative analysis of live-cell imaging are available in Supplementary Software 1.

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

## Acknowledgements

We thank all members of the Sherratt lab. for useful discussions, the departmental proteomics unit for proteomics support, and Rachel Baker for excellent technical support. We thank Frank Bürmann, Jan Löwe (MRC LMB, Cambridge, UK), and Mike Burkart (UCSD, San Diego) for helpful discussions. This work was supported by a Wellcome Investigator Award [200782/Z/16/Z to D.J.S]. MRC Programme Grants [MR/N020413/1 and MR/V028839/1] awarded to C.V.R. supported the native mass spectrometry. Funding for open access charge: Wellcome Trust [200782/Z/16/Z].

## Author contributions

J.P.P., L.K.A., and D.J.S. conceived and directed the project. J.P.P., G.L.M.F., and J.R.B. undertook biochemical experiments. M.F. undertook proteomics and analysis. J.M. helped with quantitative imaging analysis. C.V.R. provided facilities for nMS. The paper was drafted by J.P.P., L.K.A., and D.J.S., with all authors participating in the final manuscript.

## Competing interests

The authors declare no competing interests.
