## [Peer Review File · Nature Communications]

Acyl Carrier Protein promotes MukBEF action in Escherichia coli chromosome organization-segregationReviewers' Comments:

Reviewer #1:

Remarks to the Author:

Key Results: The authors present a compelling exploration of the Structural Maintenance of Chromosomes (SMC) complex formed by MukB, MukE, and MukF. They perform extensive characterization to explore several previous reports of AcpP co-purifying with MukB. They demonstrate that AcpP interacts with MukB's coiled-coils, that it is required for MukB's ATPase activity, and that its binding to MukB's joint inhibits further complexation. All of this is summarized by the conclusion "that AcpP is an essential partner in MukBEF action in chromosome organization-segregation". The authors utilized classical biochemical techniques, guided by crosslinking and native mass spectrometry to support these conclusions.

Validity: On the whole, the core conclusions are supported by the data presented. However, my overall assessment of this work is very mixed.

Data & methodology: The use of MukBN and MukBHN is convincing in localizing the interaction of AcpP with MukB. The subsequent crosslinking data and the mutation studies that resulted do seem to prove a specific interaction site. Crosslinking, while more commonplace these days, is still a very complex experiment and the methods/results should be expanded in supplemental significantly to allow the reader to judge the quality of the crosslinks identified. That being said, the crosslinking data was only used to guide subsequent mutagenesis, so the validity of those data don't really impact the findings of the mutagenesis, which the authors rely upon to support the core conclusions.

The extinction of AcpP binding was only seen in the double and triple charge inversion mutants, which may be the result not of loss of specific interactions with those residues, but conformational changes in that region. In a similar paper that did not rely on mass spectrometry, this critique would be solved by the addition of a sentence that clearly stated an overall conformational change could be an alternative explanation. Since the authors were utilizing advanced mass spectrometry techniques, it seems that simple collisional cross-section measurements would at least rule out gross changes in solution-phase structure. Alternatively, further crosslinking experiments on the mutants could reveal these changes as well if they exist through varied intramolecular links. The authors should perform additional experiments to rule out altered confirmation or alert the reader to this possibility.

The next finding relates to the requirement of AcpP for MukB ATPase activity in vitro. The ATPase activity assay results are convincing but the native mass spectrometry creates many questions. Native mass spectrometry of very large complexes can lead to unexplained mass shifts due to inadequate desolvation and adduction. However, at the molecular weights for all species displayed in figure 1, the mass accuracy should be significantly better than that displayed. In fact, it appears that the apo- and holo-AcpP signals are isotopically resolved, yet the mass differences to theoretical are ~18 Da. This likely indicates a simple error in calculating the theoretical mass (missed a water somewhere) which should be addressed. In the very same spectrum, the mass difference on MukBH is ~-118 Da. This is a ~-4,000 ppm error on an instrument capable of <5 ppm measurements in this m/z range and presents a much more troubling deviation, especially because it is negative and so cannot be explained by adduction. Can the authors explain this? This continues with MukBN-AcpP measuring 1,532 Da off of theoretical and the dimer is in error by more than twice this (3,239 Da). These mass errors create questions about the reagents that underly this work.

Continuing to figure 3, in panel a, the authors highlight two different distributions in the 1 μ M spectrum. The size of the figure makes it nearly impossible to see that these distributions are resolved. Additionally, the mass errors go from negative to positive with the only putative addition being an additional AcpP. This could be a result of the significantly overlapping peaks pushing the centroids away from the region of overlap, but this should be explored and the spectra enlarged for clarity. Perhaps an overlay with the theoretical distributions would be instructive.

This work is done extensively with recombinant proteins, so the contents of these samples are highly defined. Therefore, the species as labelled are likely the correct stoichiometry but it certainly does appear that the recombinant proteins are not quite what the authors think they made. This then becomes particularly troubling when taken in combination with the crosslinking mass spectrometry. If these mass errors are due to PTM, or differences between the intended sequence and the actual sequence, it makes the presentation of more details around the crosslinking even more important. The authors utilized more advanced mass spectrometry techniques to elucidate this system, but failed to use conventional proteomics to validate their reagents. Simple intact mass spectra of each purified component used should match to theoretical within 5 ppm or have fragmentation data that verify sequence variation or PTM. While I believe the data are successful in demonstrating stoichiometry and validating interaction between the various proteins, the data are rather sloppy. The mass errors need to be explained in order for the mass spectrometry to convincingly add to the story.

Overall, it seems as if the mass spectrometry component was inserted more as an afterthought than an integral part of the work from the outset. It is because of this, that it neither really strengthens nor weakens the work. The binding site is determined by mutagenesis and the complexes formed are largely identifiable from the BN-PAGE experiments. Other mass spectrometry experiments could have much more potently aided this work.

Appropriate use of statistics and treatment of uncertainties: How are the errors on observed mass calculated for the figures? Are they an average of the charge state errors? This should be defined. pLink should be cited. The 20 ppm tolerance and 5% FDR for crosslinking data analysis should be defended. These are rather permissive. Sequence coverage of the proteins should be shown.

Clarity and Context: The clarity of the presentation needs substantial improvement. All of the figures contain text that is significantly smaller than the manuscript text and this is prior to layout. All figures need to be made more legible. Additionally, this journal intends to take readers from a diverse background into topical subject matter outside their expertise. The content of this paper does just that for this reviewer. The abstract/introduction failed to explain to me why I should care about this protein complex, beyond its contributions to "chromosome organization-segregation". What does it do, and why is understanding AcpP's role important? In other words, the significance of the work is rather hard to determine. Both native mass spectrometry of protein complexes as well as crosslinking mass spectrometry are becoming more standard tools for structural biology, and the classical biochemical experiments are just that. Therefore, without a convincing description of the significance, it is hard for this reviewer to understand why this work is deserving of being showcased in a non-topical journal.

Outside Expertise: Reviewing the introduction and the references contained therein falls outside this reviewer's expertise as does the cell phenotype data.

Reviewer #2:

Remarks to the Author:

Several reports in the literature have indicated that the *E. coli* acyl-carrier protein AcpP may co-purify with the MukBEF complex. The functional significance of the putative interaction has remained elusive. In this manuscript, Prince et al. demonstrate a specific binding of AcpP to the neck region of the MukB protein by performing pull-downs and native mass spectrometry. The authors show that AcpP strongly stimulates the ATPase activity of purified MukBEF *in vitro*. Mutants of MukB that are defective in AcpP association display aberrant cellular dynamics (similar to an ATPase defective mutant) indicating the relevance of the AcpP-MukB association for proper MukBEF function.

Altogether, these findings uncover an exciting link between chromosome segregation and fatty acid synthesis, possibly coupling an important cell cycle event to cellular metabolism. This is a very well executed and presented study with many high-quality experiments. The absence of structural information on the AcpP-MukB interface (and on the relevant MukB region) leaves some uncertainty

on the interpretation of the MukB mutations and thus on the importance of the AcpP association for MukBEF function. Nevertheless, the work provides an important step forward and a strong basis for future investigation.

Major comments

The mutants defective in AcpP interactions may have defects in addition to preventing AcpP association, as suggested by their impact on MukBEF assembly in the absence of AcpA *in vitro*. Thus, the conclusion that 'AcpP is an essential co-factor for MukBEF action in chromosome organization-segregation' is not fully justified (albeit likely). The manuscript should be carefully reworded to acknowledge other possibilities.

The extent of defects in AcpP-MukB interaction and growth phenotype correlate well only when assuming a very high concentration of AcpP in the cell. It would be helpful to extend this correlation by artificially lowering the cellular expression of AcpP to test whether this exacerbate the otherwise weak phenotype of intermediate mutants (e.g. K1114E). Similarly, does AcpP overexpression (as in Figure S4a) rescue some of the observed phenotypes?

Mutants of the bacterial Smc proteins with strongly impaired ATPase activity have only mild phenotypic consequences in *B. subtilis*. It would be interesting to obtain similar mutations in the MukB ATP binding pocket to determine whether the phenotypes are comparable to the MukB-AcpP interaction mutants (with similar ATPase activity). If this is not feasible, the authors should at least discuss their findings in the light of the Wang et al. 2018 study.

The MukBEF fluorescence localization is a good indicator for MukBEF function. However, a more direct visualization of chromosome organization-segregation (e.g. by the formation of anucleate cells) should be used to support the conclusions.

Minor points:

AcpH and SFP need to be better explained.

A recent paper showed that *E. coli* MukBEF is (partially) functional in *P. aeruginosa*. Would *P. aeruginosa* AcpP support MukBEF function? Can the authors test for MukBEF binding of *P. aeruginosa* AcpP or at least speculate on the likelihood of interaction based on the sequence conservation?

Line 44, typo: 'klesin' should be 'kleisin'

An estimate for the molar concentration of cellular AcpP (line 57, line 206, and line 237) would be useful (for interpreting the impact on ATPase Fig. 2d).

Reviewer #3:

Remarks to the Author:

MukBEF is a well established model to understand the mechanism of SMC-like complexes in bacteria, that play essential roles in chromosome organization and segregation in bacteria. Much is known about the *in vivo* stoichiometry, and *in vitro* structure of these complexes, however only a few interaction partners have been mapped and their biological consequences investigated.

This manuscript focuses on charting interactions between MukBEF and AcpP, a highly abundant factor involved in fatty acid biosynthesis. Previous reports have presented evidence for interactions between AcpP and MukB. This manuscript identifies the AcpP region interacting with MukB and explore the functional significance of these interactions.

The ms first uses biochemical assays to identify the region of MukB that interacts with AcpP, and

resorted to mass spect to identify the interaction stoichiometry. Further crosslinking combined with mass-spec identified several MukB aminoacids (e.g. K1125) involved in AcpP interactions. Mutations in these AAs lead to abolishment of interactions. Overall, these part demonstrates specific interactions between MukB and AcpP.

Next, the authors explore the functional significance of these interactions. For this, they first show that MukBEF ability to hydrolyze ATP is compromised in absence of AcpP, but can be restored with recombinant AcpP. Interactions of AcpP and MukB seem also to be important for altering intermolecular interactions in vitro.

They assess the relevance of these interactions in vivo by introducing mutations in MukB that perturbed interactions with AcpP and by monitoring the association of MukBEF foci with the origin or terminus of replication. In WT conditions, most MukBEF foci are origin-associated. They observe a mild reduction of origin association in some of the mutants. However, there is a clear shift in the axial distribution of MukBEF foci from the origin towards the terminus regions in the mutants affecting the ATP hydrolysis of MukB. The experiments are properly executed, and analyzed. Overall, the conclusions are well supported by the data presented.

REVIEWER COMMENTS

Reviewer #1 (Remarks to the Author):

Thank you for your constructive comments regarding both clarity and context overall, and the detailed comments on the proteomics-mass spectrometry analysis. As described below we believe we have satisfactorily addressed all concerns and provide the requested information with respect to the proteomics.

Key Results: The authors present a compelling exploration of the Structural Maintenance of Chromosomes (SMC) complex formed by MukB, MukE, and MukF. They perform extensive characterization to explore several previous reports of AcpP co-purifying with MukB. They demonstrate that AcpP interacts with MukB's coiled-coils, that it is required for MukB's ATPase activity, and that its binding to MukB's joint inhibits further complexation. All of this is summarized by the conclusion "that AcpP is an essential partner in MukBEF action in chromosome organization-segregation". The authors utilized classical biochemical techniques, guided by crosslinking and native mass spectrometry to support these conclusions.

Validity: On the whole, the core conclusions are supported by the data presented. However, my overall assessment of this work is very mixed.

Data & methodology: The use of MukBN and MukBHN is convincing in localizing the interaction of AcpP with MukB. The subsequent crosslinking data and the mutation studies that resulted do seem to prove a specific interaction site. Crosslinking, while more commonplace these days, is still a very complex experiment and the methods/results should be expanded in supplemental significantly to allow the reader to judge the quality of the crosslinks identified. That being said, the crosslinking data was only used to guide subsequent mutagenesis, so the validity of those data don't really impact the findings of the mutagenesis, which the authors rely upon to support the core conclusions.

The sample preparation and LC-MS/MS experimental conditions are provided in the material and methods section of the manuscript. Additionally, the LC-MS/MS methods and results files containing proteins, peptides and cross-linked peptides identified are accessible via Proteome Xchange with identifiers PXD026062 and PXD026017, using reviewers accounts, PXD026062 : username: reviewer_pxd026062@ebi.ac.uk / password :XH6iB5X1 and PXD026017, username :reviewer_pxd026017@ebi.ac.uk / password : tDL3BTAu. Furthermore, in the revised manuscript we provide a corresponding list of MS/MS spectra in Supplementary Fig. 7.

The extinction of AcpP binding was only seen in the double and triple charge inversion mutants, which may be the result not of loss of specific interactions with those residues, but conformational changes in that region. In a similar paper that did not rely on mass spectrometry, this critique would be solved by the addition of a sentence that clearly stated an overall conformational change could be an alternative explanation. Since the authors were utilizing advanced mass spectrometry techniques, it seems that simple collisional cross-section measurements would at least rule out gross changes in solution-phase structure. Alternatively, further crosslinking experiments on the mutants could reveal these changes as well if they exist through varied intramolecular links. The authors should perform additional experiments to rule out altered confirmation or alert the reader to this possibility. We can no longer undertake further experiments, because DJS has retired and his research group dis-banded. As proposed, we have alerted the reader to the possibility of an altered conformation.

The next finding relates to the requirement of AcpP for MukB ATPase activity in vitro. The ATPase activity assay results are convincing but the native mass spectrometry creates

many questions. Native mass spectrometry of very large complexes can lead to unexplained mass shifts due to inadequate desolvation and adduction. However, at the molecular weights for all species displayed in figure 1, the mass accuracy should be significantly better than that displayed. In fact, it appears that the apo- and holo-AcpP signals are isotopically resolved, yet the mass differences to theoretical are ~18 Da. This likely indicates a simple error in calculating the theoretical mass (missed a water somewhere) which should be addressed. In the very same spectrum, the mass difference on MukBH is ~-118 Da. This is a ~-4,000 ppm error on an instrument capable of <5 ppm measurements in this m/z range and presents a much more troubling deviation, especially because it is negative and so cannot be explained by adduction. Can the authors explain this?

Thank you for identifying the discrepancies in our mass calculations. We now have reassessed both the theoretical and measured masses of our reagents. Our previous calculations used a custom software and only took the most abundant peaks into consideration (for example, red arrow peaks in figure RF1 below), leading to small discrepancies between theoretical and measured masses.

In Fig. 1d a difference of ~18 Da for apo- and holo-AcpP likely reflected a sodium adduct. When calculated manually (using the base peaks indicated by green arrows), the measured masses for apo-AcpP and holo-AcpP were 9051 ± 0.3 Da and 9391 ± 0.4 Da, respectively, in good agreement with the theoretical masses (apo-AcpP 9052 Da and holo-AcpP 9393 Da).

RF1: Zoom in view of the AcpP spectrum shown in Fig. 1a. The correct measured mass is calculated using the m/z values shown with green arrows (base peaks). The well resolved additional peaks give rise to masses that correspond to multiple sodium adducts.

Likewise, the recalculated experimental mass for MukB^H was 57512 ± 0.3 Da giving rise to a mass difference of ~-131 Da, which we believe represents N-terminal Met excision (a common modification)¹.

This continues with MukBN-AcpP measuring 1,532 Da off of theoretical and the dimer is in error by more than twice this (3,239 Da). These mass errors create questions about the reagents that underly this work.

For MukB^N we identified an error in the theoretical mass. This protein should have a mass of 41478 Da, which when in complex with AcpP should give a theoretical mass of 50288 Da or 100575 Da for a MukB^N₂-AcpP₂ dimer. This presents a difference of -31 and +144 Da to that measured. We note that this sample of MukB^N, now displayed as Fig. 1e in the revised manuscript, was in complex with native AcpP that can possess multiple modifications

because of its role in fatty acid synthesis. For theoretical mass calculations of MukB truncated mutants (MukB^N and MukB^{HN}) with native AcpP, an average of *apo*- and *holo*-AcpP was used.

Continuing to figure 3, in panel a, the authors highlight two different distributions in the 1 μ M spectrum. The size of the figure makes it nearly impossible to see that these distributions are resolved. Additionally, the mass errors go from negative to positive with the only putative addition being an additional AcpP. This could be a result of the significantly overlapping peaks pushing the centroids away from the region of overlap, but this should be explored and the spectra enlarged for clarity. Perhaps an overlay with the theoretical distributions would be instructive.

Thank you for raising this issue. As suggested, we now have enlarged the spectra for clarity and overlaid with the theoretical distributions. This is incorporated into Fig. 3b as an insert.

This work is done extensively with recombinant proteins, so the contents of these samples are highly defined. Therefore, the species as labelled are likely the correct stoichiometry but it certainly does appear that the recombinant proteins are not quite what the authors think they made. This then becomes particularly troubling when taken in combination with the crosslinking mass spectrometry. If these mass errors are due to PTM, or differences between the intended sequence and the actual sequence, it makes the presentation of more details around the crosslinking even more important. The authors utilized more advanced mass spectrometry techniques to elucidate this system, but failed to use conventional proteomics to validate their reagents. Simple intact mass spectra of each purified component used should match to theoretical within 5 ppm or have fragmentation data that verify sequence variation or PTM. While I believe the data are successful in demonstrating stoichiometry and validating interaction between the various proteins, the data are rather sloppy. The mass errors need to be explained in order for the mass spectrometry to convincingly add to the story.

We have provided explanations/corrections in the revised manuscript. Our recalculated experimental masses agree well with the theoretical masses of the reagents used.

Overall, it seems as if the mass spectrometry component was inserted more as an afterthought than an integral part of the work from the outset. It is because of this, that it neither really strengthens nor weakens the work. The binding site is determined by mutagenesis and the complexes formed are largely identifiable from the BN-PAGE experiments. Other mass spectrometry experiments could have much more potently aided this work.

The nMS was an integral part of the work from the outset and has played an important role in validating the stoichiometry of the complexes, as the reviewer accepts. Our initial objective was to determine if the AcpP interaction with MukB was specific and whether it was functionally important. Both of these objectives have been met in the manuscript. In addition, XL-MS has been extremely useful in probing the interaction site of AcpP to the neck region of MukB.

Appropriate use of statistics and treatment of uncertainties: How are the errors on observed mass calculated for the figures? Are they an average of the charge state errors? This should be defined.

For native MS data, errors on observed masses were calculated using a previously published method². The errors were an average of charge state errors. This new reference has been added to the methods section.

pLink should be cited. The 20 ppm tolerance and 5% FDR for crosslinking data analysis should be defended. These are rather permissive. Sequence coverage of the proteins should be shown.

The sequence coverage of proteins identified in in-gel and in-solution cross-linked samples are now shown in Supplementary Fig. 6. The 20 ppm tolerance and 5% FDR used for crosslinking data analysis were set as default settings in pLink2. We kept them as such while manually validating the MS/MS spectra containing cross-linked peptides of interest. As mentioned above, we now provide a corresponding list of MS/MS spectra in Supplementary Fig. 7.

Clarity and Context: The clarity of the presentation needs substantial improvement. All of the figures contain text that is significantly smaller than the manuscript text and this is prior to layout. All figures need to be made more legible.

We have attended to the above.

Additionally, this journal intends to take readers from a diverse background into topical subject matter outside their expertise. The content of this paper does just that for this reviewer. The abstract/introduction failed to explain to me why I should care about this protein complex, beyond its contributions to “chromosome organization-segregation”. What does it do, and why is understanding AcpP’s role important? In other words, the significance of the work is rather hard to determine. Both native mass spectrometry of protein complexes as well as crosslinking mass spectrometry are becoming more standard tools for structural biology, and the classical biochemical experiments are just that. Therefore, without a convincing description of the significance, it is hard for this reviewer to understand why this work is deserving of being showcased in a non-topical journal.

In the revised manuscript, we have improved both figures (and text) to improve presentation and context. We appreciate that this reviewer’s expertise is outside of the chromosome organisation-segregation field and have now endeavoured to make the context of the work more accessible to the more general reader.

Outside Expertise: Reviewing the introduction and the references contained therein falls outside this reviewer’s expertise as does the cell phenotype data.

Reviewer #2 (Remarks to the Author):

Thank you for your constructive and overall positive comments! Since this work was submitted, a bioRxiv paper showing (amongst other things) AcpP bound to MukBEF in a cryoEM structure has been put online³. This structure confirms the data and conclusions here, but provides no further insight as to the role of AcpP. This paper is now cited and discussed briefly in the Discussion.

Several reports in the literature have indicated that the E. coli acyl-carrier protein AcpP may co-purify with the MukBEF complex. The functional significance of the putative interaction has remained elusive. In this manuscript, Prince et al. demonstrate a specific binding of AcpP to the neck region of the MukB protein by performing pull-downs and native mass spectrometry. The authors show that AcpP strongly stimulates the ATPase activity of purified MukBEF in vitro. Mutants of MukB that are defective in AcpP association display aberrant cellular dynamics (similar to an ATPase defective mutant) indicating the relevance of the AcpP-MukB association for proper MukBEF function.

Altogether, these findings uncover an exciting link between chromosome segregation and fatty acid synthesis, possibly coupling an important cell cycle event to cellular metabolism. This is a very well executed and presented study with many high-quality experiments. The absence of structural information on the AcpP-MukB interface (and on the relevant MukB region) leaves some uncertainty on the interpretation of the MukB mutations and thus on the importance of the AcpP association for MukBEF function. Nevertheless, the work provides an important step forward and a strong basis for future investigation.

Major comments

The mutants defective in AcpP interactions may have defects in addition to preventing AcpP association, as suggested by their impact on MukBEF assembly in the absence of AcpA *in vitro*. Thus, the conclusion that 'AcpP is an essential co-factor for MukBEF action in chromosome organization-segregation' is not fully justified (albeit likely). The manuscript should be carefully reworded to acknowledge other possibilities.

The revised manuscript now acknowledges other possibilities, as suggested.

The extent of defects in AcpP-MukB interaction and growth phenotype correlate well only when assuming a very high concentration of AcpP in the cell. It would be helpful to extend this correlation by artificially lowering the cellular expression of AcpP to test whether this exacerbate the otherwise weak phenotype of intermediate mutants (e.g. K1114E). Similarly, does AcpP overexpression (as in Figure S4a) rescue some of the observed phenotypes?

We can no longer undertake further experiments, because DJS has retired and his research group dis-banded. We did consider lowering the cellular concentration of AcpP (using a degron, or by *ts* mutants), but were concerned that this was highly likely to have other effects on fatty acid metabolism and overall cellular physiology, thereby confounding an incisive interpretation. Similarly, the likely interpretation of any AcpP over-production experiments would likely be inconclusive; it has also been reported that high level of AcpP over-expression can inhibit cell growth⁴. Furthermore, in these types of experiment we would not know what the relative abundance of the different AcpP forms would be. In our opinion, the data presented in Fig. 2d (now Fig. 2c in the revised manuscript), showing the response of MukB ATPase to *in vitro* concentration of AcpP (through a concentration range comparable to those *in vivo*) is more informative than any that could be extracted from modulation of *in vivo* AcpP levels. The revised manuscript makes a stronger link between *in vitro* concentrations and likely *in vivo* concentration using the best and most recent experimental estimates.

Mutants of the bacterial Smc proteins with strongly impaired ATPase activity have only mild phenotypic consequences in *B. subtilis*. It would be interesting to obtain similar mutations in the MukB ATP binding pocket to determine whether the phenotypes are comparable to the MukB-AcpP interaction mutants (with similar ATPase activity). If this is not feasible, the authors should at least discuss their findings in the light of the Wang et al. 2018 study.

As stated above, we can no longer undertake further experiments. We now discuss our results in terms of the Wang 2018 study in *B. subtilis*. Parenthetically, in *E. coli* we do not have available to us the type of assay in *B. subtilis* in which phased action of Smc can be assessed, because MukBEF loads randomly onto chromosomes rather than exhibiting *ori*-specific loading as in *B. subtilis*.

The MukBEF fluorescence localization is a good indicator for MukBEF function. However, a more direct visualization of chromosome organization-segregation (e.g. by the formation of anucleate cells) should be used to support the conclusions.

We are of the opinion that MukBEF fluorescence analysis is the best indicator of MukBEF function; particularly when we relate the position of MukBEF foci to *ori* and/or *ter*.

Nevertheless, we have extended analysis of anucleate cell formation to mutants deficient in AcpP binding (Supplementary Fig. 5b). Briefly, the triple MukB^{KRK} mutant forms anucleate cells at a frequency commiserate with its temperature-sensitivity and abnormal positioned foci, whereas the single and double mutants form anucleate cells at ~WT frequency.

Minor points:

AcpH and SFP need to be better explained.

In the revised manuscript we have provided a more detailed explanation of AcpH and SFP.

A recent paper showed that *E. coli* MukBEF is (partially) functional in *P. aeruginosa*. Would

P. aeruginosa AcpP support MukBEF function? Can the authors test for MukBEF binding of *P. aeruginosa* AcpP or at least speculate on the likelihood of interaction based on the sequence conservation?

For the reasons stated above, we can no longer undertake further experiments, however, an alignment of *E. coli* and *P. aeruginosa* AcpP sequences revealed the two to be 87% identical. Furthermore, a recent cryo-EM structure of *P. thracensis* MukBEF revealed it to stably interact with *E. coli* AcpP³. *P. thracensis* and *E. coli* AcpP sequences are 85% identical. Given the high level of sequence conservation in AcpP, we surmise that *P. aeruginosa* AcpP would be able to interact with and promote activity in *E. coli* MukBEF.

Line 44, typo: 'klesin' should be 'kleisin'

Attended to.

An estimate for the molar concentration of cellular AcpP (line 57, line 206, and line 237) would be useful (for interpreting the impact on ATPase Fig. 2d).

Attended to—we have added an estimated *in vivo* concentration as appropriate. We have assumed (as we now state) that 1 molecule/average minimal grown cell (volume ~1 micron³) equates to ~1nM.

Reviewer #3 (Remarks to the Author):

Thank you for your very positive comments!

MukBEF is a well established model to understand the mechanism of SMC-like complexes in bacteria, that play essential roles in chromosome organization and segregation in bacteria. Much is known about the *in vivo* stoichiometry, and *in vitro* structure of these complexes, however only a few interaction partners have been mapped and their biological consequences investigated.

This manuscript focuses on charting interactions between MukBEF and AcpP, a highly abundant factor involved in fatty acid biosynthesis. Previous reports have presented evidence for interactions between AcpP and MukB. This manuscript identifies the AcpP region interacting with MukB and explore the functional significance of these interactions.

The ms first uses biochemical assays to identify the region of MukB that interacts with AcpP, and resorted to mass spect to identify the interaction stoichiometry. Further crosslinking combined with mass-spec identified several MukB aminoacids (e.g. K1125) involved in AcpP interactions. Mutations in these AAs lead to abolishment of interactions. Overall, these part demonstrates specific interactions between MukB and AcpP.

Next, the authors explore the functional significance of these interactions. For this, they first show that MukBEF ability to hydrolyze ATP is compromised in absence of AcpP, but can be restored with recombinant AcpP. Interactions of AcpP and MukB seem also to be important for altering intermolecular interactions *in vitro*.

They assess the relevance of these interactions *in vivo* by introducing mutations in MukB that perturbed interactions with AcpP and by monitoring the association of MukBEF foci with the origin or terminus of replication. In WT conditions, most MukBEF foci are origin-associated. They observe a mild reduction of origin association in some of the mutants. However, there is a clear shift in the axial distribution of MukBEF foci from the origin towards the terminus regions in the mutants affecting the ATP hydrolysis of MukB. The experiments are properly executed, and analyzed. Overall, the conclusions are well supported by the data presented.

References

- 1 Hirel, P. H., Schmitter, M. J., Dessen, P., Fayat, G. & Blanquet, S. Extent of N-terminal methionine excision from Escherichia coli proteins is governed by the side-chain length of the penultimate amino acid. *Proc. Natl. Acad. Sci. U.S.A.* **86**, 8247-8251, (1989).
- 2 McKay, A. R., Ruotolo, B. T., Ilag, L. L. & Robinson, C. V. Mass Measurements of Increased Accuracy Resolve Heterogeneous Populations of Intact Ribosomes. *J. Am. Chem.Soc.* **128**, 11433-11442, (2006).
- 3 Bürmann, F., Funke, L. F. H., Chin, J. W. & Löwe, J. Cryo-EM structure of MukBEF reveals DNA loop entrapment at chromosomal unloading sites. *bioRxiv*, 2021.2006.2029.450292, (2021).
- 4 Keating, D. H., Carey, M. R. & Cronan, J. E. The Unmodified (Apo) Form of Escherichia coli Acyl Carrier Protein Is a Potent Inhibitor of Cell Growth (*). *J. Biol.Chem.* **270**, 22229-22235, (1995).

Reviewers' Comments:

Reviewer #1:

Remarks to the Author:

The authors adequately responded to the issues raised during initial review.